

# Comparison of body pressure distribution in healthy subjects between bubble wrap and an emergency mattress laid on a cardboard bed: a randomized controlled crossover trial

Seiji Hamanishi, Yukiko Asada, Yu Ikushima, Yurika Ikeda and Mai Chinushi

Nursing Faculty, Kansai University of Social Welfare, Ako, Hyogo, Japan

Corresponding author
Seiji Hamanishi,
sj.eyes76@gmail.com

## ABSTRACT

**Background:** It has been pointed out that the poor environment of evacuation shelters causes health problems and disaster-related deaths among evacuees, and we are concerned that their environment will deteriorate, particularly during a large-scale disaster due to a shortage of daily necessities. In Japan, evacuees usually slept on floors with futons until the Great East Japan Earthquake, but cardboard beds were installed in evacuation shelters. Previous studies have suggested that cardboard beds can reduce cold air transmission from the floor. We have reported that a cardboard bed can have a low-contact pressure dispersion capacity and cannot reduce musculoskeletal strain, unlike a futon or mattress. In the Great East Japan Earthquake, 33% of disaster-related deaths were reported to have been caused by physical or mental fatigue due to living in evacuation shelters. When a large-scale disaster such as the Nankai Trough Earthquake generates huge numbers of evacuees, the supply of mattresses for evacuees will be very difficult. Therefore, we considered potential alternatives that could be produced in large quantities over a short period. Bubble wrap, with very lightweight and waterproofing, could be a good candidate for mattress replacement. This study aimed to investigate the improvement in body pressure distribution and pressure-sensing area when using bubble wrap.
**Methods:** Twenty-seven healthy subjects allocated to sequences A and B with different intervention order were laid in supine and lateral positions on a cardboard bed without a mattress, bubble wrap, or air mattress: the mattress-body contact pressure and contour areas were measured, and subjective firmness and comfort during these conditions were also investigated using the visual analog scale (VAS). Acquired data were analyzed using a linear mixed-effects model and Bonferroni's *post-hoc* test, and $P < 0.05$ was considered statistically significant.
**Results:** The mattress-body contact pressure and contour area showed significant differences with and without air mattresses. With the air mattresses, the pressure in the supine position decreased by 34%, and that in the lateral position decreased by 13%. However, the four-fold bubble wrap did not improve the mattress-body contact pressure and contour area; the change ratios were within 5% compared to the cardboard bed. However, there were significant differences in subjective firmness and comfort using the VAS among all experimental positions.

**Conclusion:** Our study showed that bubble wrap could not significantly improve body pressure concentration and may not be a satisfactory substitute for air mattresses. Because of the improvement in subjective firmness and comfort with the bubble wrap, using it for an extended period may affect the incidence of back pain in evacuees. Finally, we hypothesize that the body pressure dispersion of the bubble wrap may be improved by changing the air-filling rate and the size of the air bubbles.

## INTRODUCTION

It has been pointed out that the poor environment of evacuation shelters causes health problems and disaster-related deaths among evacuees, and we are concerned that their environment will deteriorate, particularly during a large-scale disaster such as a Nankai Trough earthquake due to a shortage of daily necessities (*Ichiseki, 2013*; *Tsuboi et al., 2022*). According to the third edition of the Nankai Trough Earthquake Countermeasures Plan, approximately 4.4–9.5 million people are expected to be evacuated 1 week after an earthquake, which is 20 times the number of evacuees in the Great East Japan Earthquake (*Cabinet Office of the Government of Japan, 2013*). Physical and mental fatigue caused by living in evacuation shelters accounted for 33% of disaster-related deaths in the Great East Japan Earthquake, and it has been reported that 20% of evacuees complained of back pain (*Ichiseki, 2013*; *Tsuboi et al., 2022*). Japanese evacuees were used to sleeping directly on the floors of school classrooms or gymnasiums, but cardboard beds have been installed since the Great East Japan Earthquake (*Nara et al., 2013*). A survey in the Ishinomaki area damaged by the Great East Japan Earthquake reported that installing cardboard beds improved evacuees' health problems, such as insomnia and back pain (*Nara et al., 2013*). Other studies have suggested that the height of the cardboard beds is useful for intercepting cold air transmission from the floor (*Okamoto-Mizuno et al., 2016*). However, despite being an important indicator of back pain risk, the body pressure dispersion capacity of cardboard beds had not been elucidated. Therefore, we investigated this question and found no difference in body pressure distribution between a cardboard bed and a blue sheet on the floor. Further, we found that using cardboard with a blanket may not contribute to preventing back pain in evacuees (*Hamanishi, 2021*). Therefore, many evacuees usually use a donated futon or any mattress by laying them on a cardboard bed.

According to the Guidelines for the Operation of Evacuation Shelters, Japanese municipalities had not been required to stockpile bedding without blankets, and cots and mattresses, including futons, are to be considered for installation 72 h after the disaster (*Cabinet Office of the Government of Japan, 2016*). Therefore, many municipalities have concluded agreements with the cardboard industry group to provide cardboard beds in the event of large-scale disasters. However, when the Kumamoto Earthquake occurred in 2016, more than 60% of municipalities responded that enough cardboard beds could not be installed in evacuation shelters, despite the agreements (*Sueta et al., 2019*). In the case of a

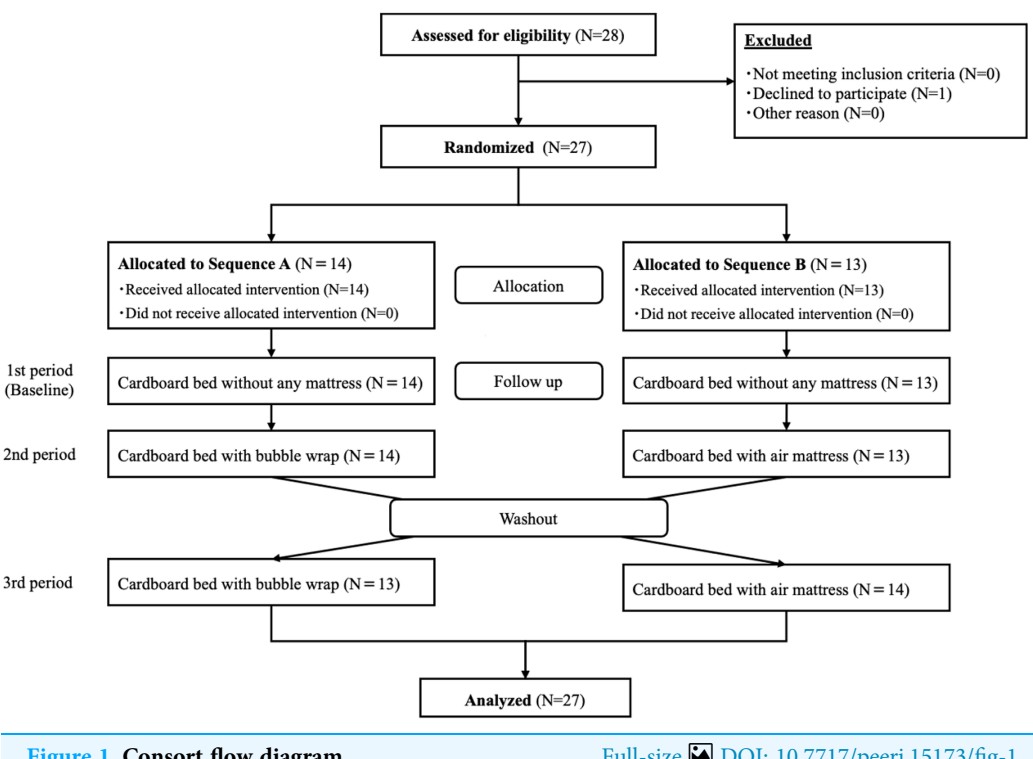

**Figure 1 Consort flow diagram.**

Nankai Trough earthquake, which is expected to affect vast areas of Japan, causing extensive infrastructure damage, it is even more difficult to provide bedding, such as cardboard beds. Furthermore, it is estimated that even blankets stockpiled by municipalities will be in short supply, and that Japan does not have a provisioning system for futons or mattresses in large quantities, such as cardboard beds. This insufficient provision system may cause health problems in evacuees, such as back pain or sleep disorders. However, to our knowledge, the effective prevention of these problems in large-scale disasters has not yet been established.

Therefore, we considered potential alternatives that could be produced in large quantities over a short period. Bubble wrap with a very lightweight and waterproofing could be a good candidate for mattress replacement. This study aimed to investigate the improvement in body pressure distribution and pressure-sensing contour area when using bubble wrap. Since a cardboard bed is known to reduce dust inhalation and interrupt the coldness transmitted from the floor, it is expected to contribute more than ever to maintaining the health of evacuees if the body pressure concentration can be improved.

# MATERIALS AND METHODS

## Study subjects

Healthy subjects were recruited using our website from. As shown in the flowchart in Fig. 1, one person was excluded because she declined to participate. The selection criteria were as follows. The inclusion criteria were: (1) age $\geq$ 18 and $\leq$ 60 years; and (2) ability to provide informed consent. The exclusion criteria were: (1) presence of musculoskeletal

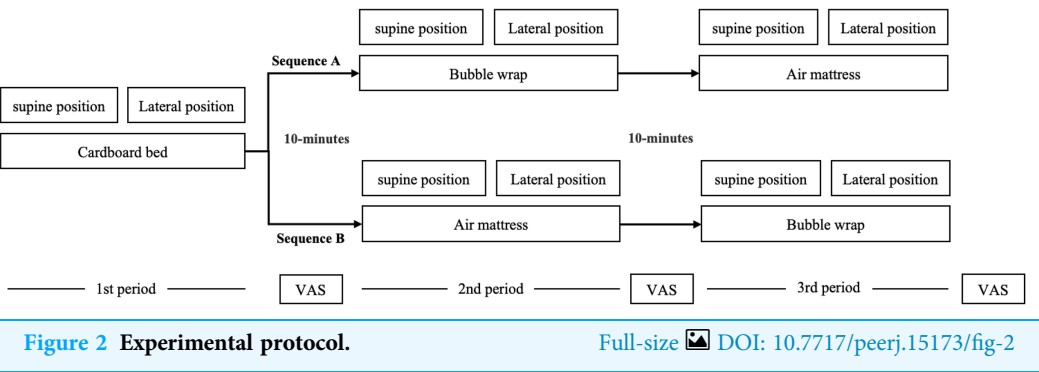

**Figure 2  Experimental protocol.**                 

pain or neuralgia; (2) difficulty in assuming supine and lateral positions due to pain; and (3) people with a disease that may change suddenly. The minimum sample size for this study was calculated to be 24 using a power analysis for mean body pressure, specifying repeated measures analysis of variance (ANOVA), using conventional effect size = 0.4, $\alpha = 0.05$, and $\beta = 0.80$ (G*power, Heinrich-Heine-Universität Düsseldorf, Düsseldorf, Germany). Therefore, the target sample size was determined to be between 24 and 30 participants. Eventually, 27 people participated in the study.

## Design and protocol

The study protocol is illustrated in Fig. 2. This was a randomized controlled crossover study, and data were acquired at Kansai University of Social Welfare in September 2022. All subjects consented to participate in the study and were randomly assigned to sequence A ($n = 14$) and B ($n = 13$) using a random number calculated in Microsoft Excel (Microsoft, Redmond, WA, USA). The mattress conditions for subjects allocated to Sequence A were measured in the following order: 1st: Cardboard bed without mattress (None); 2nd: Four-folded bubble wrap; 3rd: Air mattress. The bed conditions for subjects allocated to Sequence B were measured in the following order: 1st: Cardboard bed without mattress (None); 2nd: Air mattress; 3rd: Four-folded bubble wrap. A 10-min washout period was inserted between each experimental period. Since body pressure distribution was measured twice for each mattress condition, subjects were asked to repeat the same position twice in a row. The time (10 s per single measurement) was taken to stabilize the value before each measurement.

## Mattress

As shown in Fig. 3, the experimental conditions in this study were set as (A) a cardboard bed without any mattress (None), (B) a cardboard bed with an emergency air mattress, and (C) a cardboard bed with a four-folded bubble wrap. All subjects were required to lie in supine and lateral positions on each mattress. The thickness of the air mattress and the four-folded bubble wrap was 50 and 12 mm, respectively (the height of the particles: 3 mm).

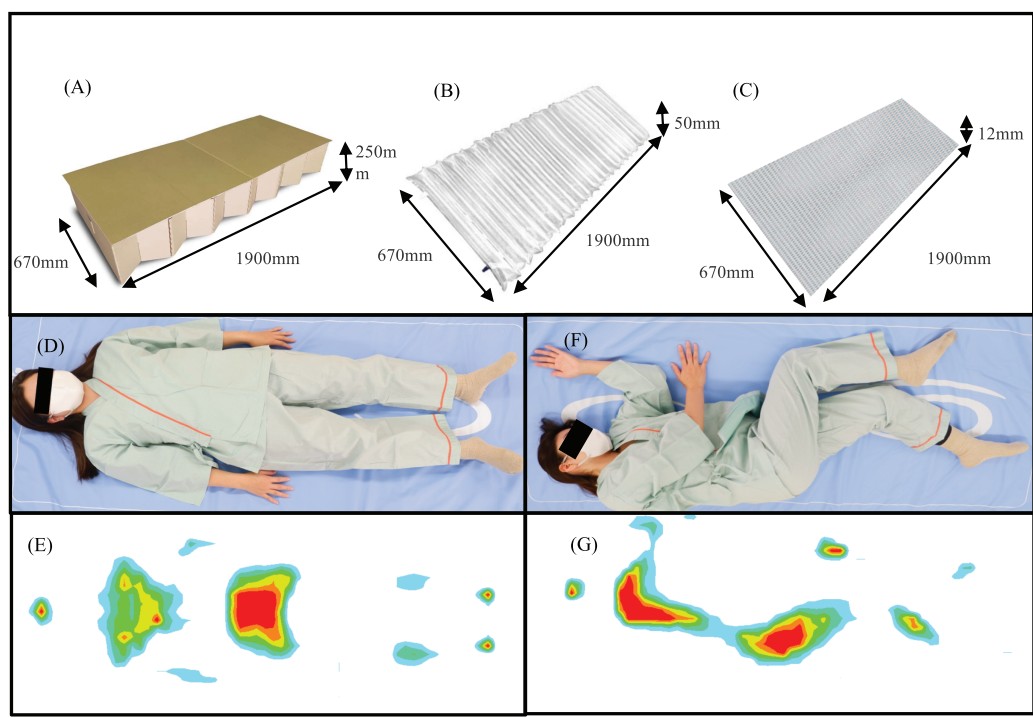

**Figure 3 Illustration of the experimental conditions.** (A) Cardboard bed. (B) Air mattress. (C) Bubble wrap. (D and E) Spine position. (F and G) Lateral position.

## Body pressure distributions

SR Soft Vision (Sumitomo Science & Engineering, Chiba, Japan) was the body pressure measurement system (BPMS), and the device was arranged with a total of 1,600 (64 × 25) pressure sensors. Each sensor size is 784 mm² (28 × 28 mm), and all sensing area of the device is 1,800 × 700 mm. The body-mattress contact pressure and body contour area of the whole body in the supine and lateral positions were measured. Each subject wore the same clothes during all experiments to minimize variations due to clothing. Further, the detection range of contact pressure was set to 10–110 mmHg to avoid detecting the pressure of the clothes. BPMS recorded pressures to 110 mmHg, so actual pressures above this threshold were recorded as 110 mmHg. To stabilize the measured values, the contact pressure distribution of the model was measured after approximately 10 s (*Hamanishi, 2021*). The measurement error (Max − Min/Mean × 100%) was 2.7% when the pressure sensing area was measured 10 times using a dummy model before the experiment. In addition, the manufacturer verified this device's accuracy in January 2022.

## Questionnaire

We collected patient information on individual attributes, including age, sex, and body mass index (BMI). In addition, subjective sleep comfort and bed surface firmness were confirmed using a visual analog scale (VAS). Two types of VAS ranges were set as 0 (most comfortable)–10 (worst possible uncomfortable) and 0 (most possible softness)–10 (worst possible firmness). Subjects spent approximately 20 s on the bed and evaluated the perceived firmness and comfort of the bed during this time.

## Data analysis

We used a linear mixed-effects model to examine the effect of bubble wrap on body pressure distribution. Period, sequence, and mattress were entered into the model as fixed effects, and subjects nested within the sequence were entered as random effects. A Bonferroni correction was performed to compare the main effects among different mattress conditions. Age, sex, and BMI were entered into the model as covariates to control for potential confounding factors. In our study, a $P$-value < 0.05 was considered statistically significant. All data were compiled using Microsoft Excel (Office 365; Microsoft, Redmond, WA, USA) and analyzed statistically using IBM SPSS (version 28.0; SPSS, Chicago, IL, USA).

## Ethics

All participants provided written informed consent prior to participation. The study procedures were conducted in accordance with the approval granted by the Ethical review board of Kansai University of Social Welfare (No. 4—0801). This study was registered in the University Hospital Medical Information Network Clinical Trials Registry (No. UMIN000048145).

## RESULTS

The participant characteristics are shown in Table 1. The participants were predominantly female (89.9%), with a mean age of 21.5 years and a BMI of 21.1 kg/m$^2$.

Figure 4 presents the results of a linear mixed model for repeated ANOVA to compare the body-mattress contact pressure distribution or VAS related to the subjective firmness and comfort among different mattress conditions. All results relative to the contact pressure and contour area showed significant differences between air mattresses and a cardboard bed without a mattress (None) but not between four-folded bubble wrap and no mattress. Using an air mattress decreased the contact pressure in the supine position by 34% and 13% in the lateral position. The contour areas of the supine and lateral positions when using an air mattress were approximately 34% and 33% larger than that of a cardboard bed without a mattress, respectively. However, the change ratio for the body-mattress contact pressure and the contour area was less than 5% compared to the cardboard bed when using bubble wrap. In addition, there were significant differences in subjective firmness and comfort using the VAS among all the experimental conditions. "Sequence" entered in our model as a fixed effect was not significant, and "Period" was not also significant between the 2nd and 3rd periods.

## DISCUSSION

In this study, we found that bubble wrap could not improve body pressure concentration, as with an air mattress, when using a cardboard bed at an evacuation shelter. However, subjective firmness and comfort were improved using the bubble wrap.

No cot had been installed in Japanese evacuation shelters, and evacuees had to sleep on the floor with a futon or mattress until the Great East Japan earthquake. Since then, cardboard beds have been installed in evacuation shelters to protect evacuees from various

**Table 1 Characteristics of participants.**

| | Sequence A | Sequence B |
|---|---|---|
| Sex | | |
| Male | 0 (0%) | 3 (23%) |
| Female | 14 (100%) | 10 (77%) |
| Age | 21.50 ± 0.65 | 21.54 ± 0.52 |
| BMI | 21.43 ± 3.68 | 20.68 ± 2.36 |
| Mean body-pressure (mmHg) | | |
| Spine position | | |
| None | 40.43 ± 4.12 | 38.92 ± 4.14 |
| Bubble wrap | 40.11 ± 3.59 | 38.50 ± 2.83 |
| Air mattress | 30.39 ± 3.66 | 29.88 ± 2.56 |
| Lateral position | | |
| None | 41.64 ± 4.59 | 40.69 ± 3.15 |
| Bubble wrap | 39.96 ± 3.03 | 40.50 ± 2.48 |
| Air mattress | 35.86 ± 3.95 | 36.77 ± 3.59 |
| Pressure-sensing contour area (cm$^2$) | | |
| Spine position | | |
| None | 1,491.00 ± 326.72 | 1,504.07 ± 282.71 |
| Bubble wrap | 1,520.40 ± 356.60 | 1,582.17 ± 279.09 |
| Air mattress | 2,208.92 ± 330.97 | 2,264.86 ± 225.35 |
| Lateral position | | |
| None | 1,335.88 ± 252.90 | 1,345.77 ± 212.31 |
| Bubble wrap | 1,414.28 ± 246.97 | 1,426.28 ± 224.21 |
| Air mattress | 1,978.48 ± 294.00 | 1,960.91 ± 158.67 |
| Subjective firmness (VAS) | | |
| None | 7.66 ± 1.24 | 7.58 ± 1.51 |
| Bubble wrap | 5.73 ± 1.52 | 5.76 ± 1.74 |
| Air mattress | 2.39 ± 1.26 | 1.75 ± 0.80 |
| Subjective comfort (VAS) | | |
| None | 6.18 ± 1.91 | 6.29 ± 2.36 |
| Bubble wrap | 4.80 ± 1.50 | 4.56 ± 1.89 |
| Air mattress | 1.99 ± 1.36 | 2.10 ± 1.07 |

health problems when a large-scale disaster occurs. Previous studies on healthy subjects have reported that they are more likely to develop musculoskeletal pain when lying on the floor than when lying on a mattress and that body-mattress contact pressure and contour area when lying on a cardboard bed without any mattress do not differ from that when lying directly on the floor (*Lee et al., 2016*; *Hamanishi, 2021*). The present study also showed that the body pressure distribution of a cardboard bed with bubble wrap was not different from that of a cardboard bed without any mattress. These results indicate that the body-mattress contact pressure distribution when using a cardboard bed with bubble wrap did not differ from that when lying on the floor. Bubble wrap, with very lightweight and

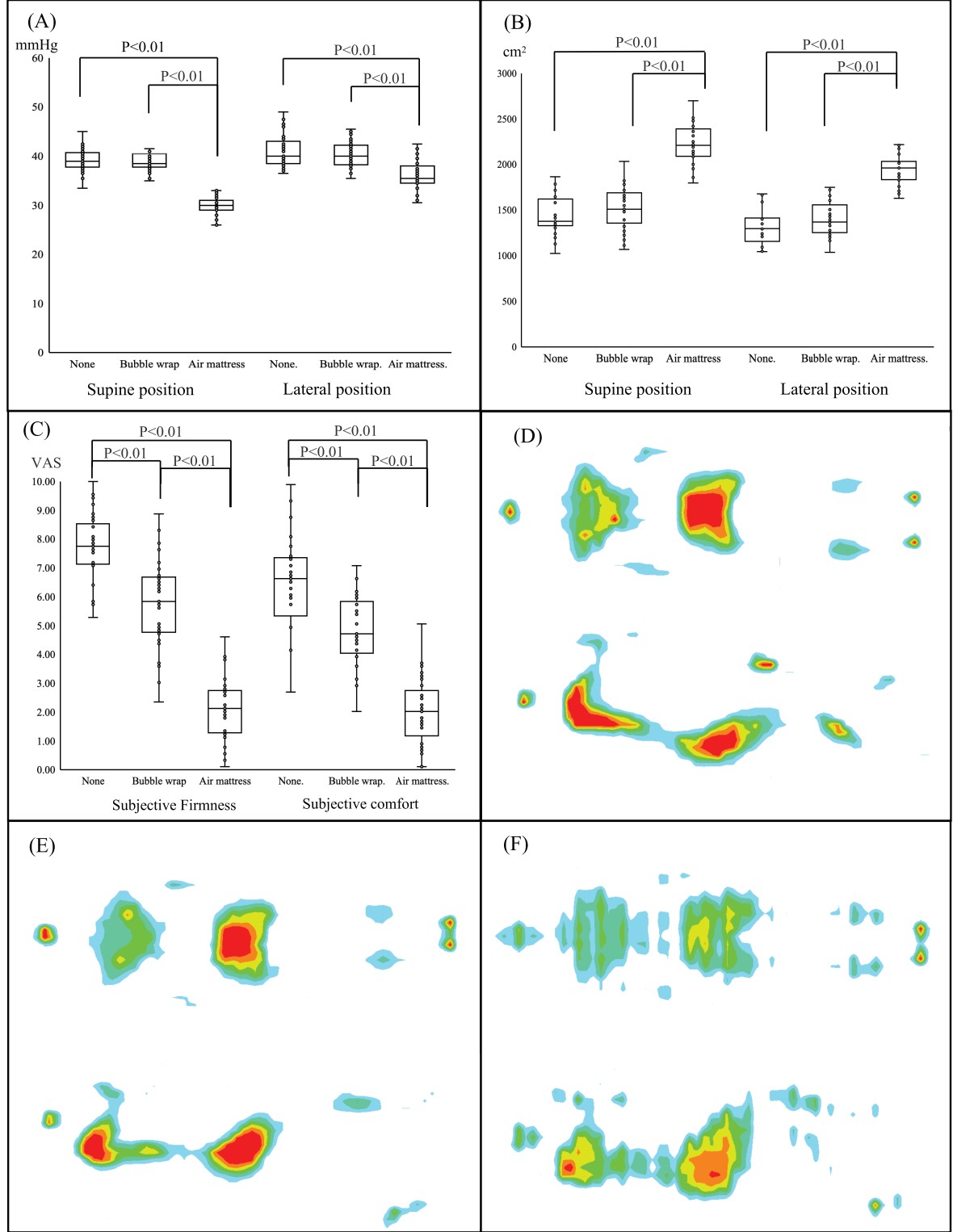

**Figure 4 Comparison of body pressure distribution among three mattresses.** Comparison of (A) body-mattress contact pressure, (B) body surface contour area, and (C) subjective firmness and comfort among the three mattresses using a linear mixed model and Bonferroni *post-hoc* test. Data are represented for box and whisker plots. The center line denotes the median value, while the box contains the 25th to 75th percentiles of dataset. The whiskers mark the minimum and maximum value. Illustration of body pressure distribution for (D) cardboard bed, (E) bubble wrap, (F) emergency air mattress.

waterproofing, can be produced in large quantities within a short period; hence, it is a good candidate for emergency mattress replacements. However, our results showed that the bubble wrap could not improve the body pressure concentration compared to an air mattress, although an emergency air mattress could improve it. In short, our results may indicate that a bubble wrap may not be a satisfactory substitute for air mattresses. Several studies have indicated that a mid-firm mattress is useful for avoiding musculoskeletal pain and improving sleep quality (*Kovacs et al., 2003*; *Normand et al., 2005*; *López-Torres et al., 2008*; *Jacobson et al., 2010*). An ideal mattress can push back against weight and hold the spine's alignment in a neutral position.

However, an excessively firm mattress could cause back pain because the weight is concentrated on the buttocks and shoulders. Although four stacked layers of bubble wrap were used in this study, they may not have been sufficient to disperse human weight because the particle size of a single bubble was only 10 mm. The amount of filled air in a single bubble also affects the rebound. Therefore, the capacity of body pressure dispersion by bubble wrap can be improved by changing the air-filling rate and the size of the air bubbles. It will be necessary to investigate how bubble wraps with different numbers of folded layers and bubble sizes will affect body pressure distribution. We will also obtain data on the effects of bubble wrap with different air-filling volumes on body pressure distribution. However, subjective firmness and comfort were improved by laying a bubble wrap on a cardboard bed. In the event of a large-scale disaster, some evacuees were forced to spend months in shelters, but the subjects were asked to lay on a bed for a very short time in this study. The amount of time spent lying down is an essential factor in influencing the strain on the musculoskeletal system. Therefore, long-term use of bubble wrap may affect the incidence of back pain in evacuees. Chronic pain, including back pain, is a known risk factor for depression (*He et al., 2022*). Evacuees are prone to mental illness due to fear, loss, and grieving for their future lives, and musculoskeletal pain caused by inadequate bedding systems can exacerbate their mental symptoms (*He et al., 2022*). Deaths due to the exacerbation of health problems, accidents, and suicides after the disaster are called disaster-related deaths. In the Kumamoto earthquake, disaster-related deaths were higher than direct disaster deaths (*Sueta et al., 2019*). Improving the shelter environment is important for maintaining the physical and psychological health of disaster victims and preventing disaster-related deaths; however, Japanese municipalities have not stockpiled bedding other than blankets. For this reason, many municipalities in Japan have signed agreements with industry associations to provide cardboard beds in the event of a disaster. However, the supply of futons and mattresses has depended on donations even now (*Japan Corrugated Case Association (JCCA), 2020*). It is more difficult than ever to provide bedding in the event of a Nankai Trough earthquake, which is expected to cause approximately 20 times as many evacuees as the Great East Japan Earthquake. Therefore, stockpiling or establishing a supply system for emergency mattresses for large-scale disasters such as the Nankai Trough earthquake could reduce evacuees' health problems. The low cost of ownership and easy storage are positive characteristics of a bubble wrap.

This study has several strengths. First, to our knowledge, this randomized controlled crossover trial is the first study aiming to improve evacuees' body pressure concentration

in a shelter environment where futons or any mattresses laid on a cardboard bed cannot be provided. Second, our results suggest that personal stockpiling or a building system to provide mattresses for evacuees may contribute to maintaining evacuees' health when a large-scale disaster occurs.

However, this study had some limitations. First, obese, or older people have a higher risk of back pain, but our subjects were mostly young females who were not obese. As the results in high-risk individuals may differ from ours, we adjusted for these effects by entering sex and BMI as covariates in our linear mixed model. Second, we considered that the carryover effect of body pressure distribution was not strong; therefore, we set the washout period to only 10 min. However, sequence and period (2nd–3rd), entered as fixed effects, did not significantly affect our results. The average body contact pressure was calculated using a maximum peak pressure of 110 mmHg. Therefore, localized peak pressures exceeding this value were not recorded and the actual average pressure may be higher than shown. In this study, the same size bubbles were used without taking body shape into account. However, the appropriate height of the bubble is likely to vary depending on variables such as weight, gender, and age. Individual factors should be considered when selecting a mattress of appropriate thickness.

## CONCLUSIONS

Our study showed that bubble wrap could not significantly improve body pressure concentration and may not be a satisfactory substitute for air mattresses. Because of the improvement in subjective firmness and comfort using bubble wrap, we hypothesize that using bubble wrap for a long period may affect the incidence of back pain in evacuees.

## ACKNOWLEDGEMENTS

The author would like to thank Editage for English language editing.

### Funding
This work was supported by the Japan Society for the Promotion of Science (JSPS) KAKENHI Grant Number 20K19264. The funders had no role in study design, data collection and analysis, decision to publish, or preparation of the manuscript.

### Grant Disclosures
The following grant information was disclosed by the authors:
Japan Society for the Promotion of Science (JSPS) KAKENHI: 20K19264.

### Competing Interests
The authors declare that they have no competing interests.

## Author Contributions

- Seiji Hamanishi conceived and designed the experiments, performed the experiments, analyzed the data, prepared figures and/or tables, authored or reviewed drafts of the article, and approved the final draft.
- Yukiko Asada performed the experiments, analyzed the data, authored or reviewed drafts of the article, and approved the final draft.
- Yu Ikushima performed the experiments, analyzed the data, authored or reviewed drafts of the article, and approved the final draft.
- Yurika Ikeda performed the experiments, analyzed the data, authored or reviewed drafts of the article, and approved the final draft.
- Mai Chinushi performed the experiments, analyzed the data, authored or reviewed drafts of the article, and approved the final draft.

## Human Ethics

The following information was supplied relating to ethical approvals (*i.e.*, approving body and any reference numbers):

The Ethical review board of Kansai University of Social Welfare.

## Data Availability

The raw measurements are available in the Supplemental File.

## Clinical Trial Registration

The following information was supplied regarding Clinical Trial registration:

UMIN000048145

https://center6.umin.ac.jp/cgi-open-bin/ctr_e/ctr_view.cgi?recptno=R000054855

## Supplemental Information

Supplemental information for this article can be found online at http://dx.doi.org/10.7717/peerj.15173#supplemental-information.

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
