# Peer review of "Comparison of body pressure distribution in healthy subjects between bubble wrap and an emergency mattress laid on a cardboard bed: a randomized controlled crossover trial"

_PeerJ, doi:10.7717/peerj.15173_

## Round 0.1 · original submission · Minor Revisions

The reviews provide information on all aspects of the methods and results.

·

Basic reporting

Overall, This is a comprehensive description of a cross-over design to evaluate different support surfaces that can be used in post-disaster evaluation facilities. The overall intent is vey pragmatic, attempting to improve the conditions in a shelter. Therefore, it has important relevance to these difficult situations. The paper provides information on all aspects of the methods and results. Several items ca be made more clear for the reader, including the measurement procedures. In addition, one discrepancy is noted between a table and figure that requires attention.

Abstract. The abstract includes the appropriate information but its introduction can be improved by introducing the overall premise. It leads with a reference to a historical event which the reader might not be familiar. Moreover, the premise is related to any high evacuation event, not specific disaster events, and this can be the lead thought.

Experimental design

Design and protocol- Please describe the 3 support surfaces. While dimensional figures were included, they were not referenced when the surfaces were introduced. The thickness of the bubble wrap is not disclosed, just the cell diameter in the figure

line 144. State the measurement range of the BPMS and approximate accuracy or % error as determined after or during calibration. A mentioned of when the system was calibrated would also be helpful. Also, state the size of the sensels.

line 135. body-mattress contact pressure should be defined fully. Is this the average pressure of all values exceeding 10 mm Hg? Is 'body contour area' the area of contact represented by all sensel values >10 mmHg?

line 137 the detection range of contact pressure was set to 10-110.
The minimum value is clarified and is a valid choice to define meaningful contact. The maximum is unclear- how were values exceeding 110 mmHg managed in the calculation of contact pressure and contact area? The implication is that they were not included in analysis.

line 139. Contact pressure was measured after 10 secs. Was this a single sample or were pressures measured over the duration of each posture? if multiple samples, the sampling frequency should be stated

line 144. State the duration of time that subjects spent on the surfaces before the VAS was completed. This has a large impact on the results as time is a noted factor in assessing comfort and discomfort. Whatever the duration, it should be mentioned as a decision that impacted results

line 144. Please report the prompt for the VAS, confirm if it was a 10 point scale, and state the anchor descriptors at the max and min points

Data analysis. A mixed-effects linear model is an appropriate means to assess the variables. The description was thorough

Ethics. Human subject research ethics were described

Validity of the findings

Data is presented in a table and figure so the reader has access to both, which improved readability.

line 172. area is not defined by width alone, so the term 'wider' should not be the descriptor

line 206. The authors acknowledge that the choice of bubble wrap directly impacted the results. The This would be a time the present the option of other designs or interventions. How many folds might be needed. How would larger cell wrap work? Would ‘heavy duty bubble wrap’ be a consideration. Suggesting different internal air pressure is not feasible. Bubble wrap is made for a purpose and its design reflects that purpose. This study focuses on commercial products.

Table 1 is titled incorrectly since results are also included
Data on interface pressure in Table 1 indicates the air mattress as being markedly higher than the other conditions
Table 1 pressures should be the same as the box plot pressures

Additional comments

limitations. In addition to strengths, limitations should be mentioned. Several have been mentioned above, such as duration of maintaining the posture before measurement. the choice of bubble wrap and number of folds, the use of average pressure over localized pressure measurements...to name a few. All methodological decisions come with tradeoffs, so it is useful to acknowledge them. Researchers cannot be expected to address every scenario or option.

Reviewer 2 ·

Basic reporting

Thank you for the opportunity to review this manuscript. The authors analyzed the comfort level of 3 different mattress surfaces to be used as emergency beds for potential refugees due to a large-scale disaster (earthquake). The objective of the work is extremely interesting and of great importance to providing effective aid to those displaced by a seismic event in the immediate future, guaranteeing acceptable and compatible sleep conditions in an extreme emergency condition that the civil protection organs of all countries, with high seismic incidence, should consider. Beyond the important clinical, social, and psychological health consequences of this study, I would like to underline some minor revisions:

Experimental design

no comments

Validity of the findings

- Please add in detail the technical characteristics of the body pressure sensor device
- In the statistical analysis miss the effect size and the IC95% for each variable
- Include the reliability and validity of the measure

Additional comments

- In the discussion and in the conclusions, I suggest emphasizing the positive effects of air mattresses compared to bubble wrap.

·

Basic reporting

The publication is interesting, but lacks some important variables such as the small number of participants, age and prevalence of female sex and body weight.
Another variable to consider is the height of the bubble wrap,
article to review

Experimental design

The thickness of the bubble wrap could be increased based on the weight of the subject

Validity of the findings

no comment

Additional comments

ease of setting up the beds,
low cost of ownership,
convenience in storing the material

---

## Round 0.2 · accepted · Accept

The authors have significantly improved the initial text, responding to the required revisions. The topic is innovative and of great scientific interest, but above all, it offers support for emergency management.

·

Basic reporting

The authors made multiple edits to the manuscript in response to reviewer comments and questions. The resulting revision is more clear for readers to understand the methods, analysis and results

Experimental design

nothing new to add for this revision

Validity of the findings

nothing new to add to this revision

Additional comments

2 small issues should be considered to add clarity about interface pressure measurement. Neither in my opinion rise to the level of a re-review but may help readers better understand the article content

In methods section, the statement is worded in a manner that is not clear unless the reader is familiar with pressure measurement systems, so a slight change can be considered -
“BPMS recorded pressures to 110 mmHg so actual pressures above this threshold were recorded as 110 mmHg”.

Under limitations, the current test reads:
The average body contact pressure was calculated by assuming that the value exceeding
the detection range was 110 mmHg. Therefore, the actual average pressure may be higher than
shown.

The following statement or something similar identifies the 2 issues of using a max recorded value, namely loss of peak pressure information and potential impact on average pressure
"The average body contact pressure was calculated using a maximum peak pressure of 110 mmHg. Therefore, localized peak pressures exceeding this value were not recorded and the actual average pressure may be higher than shown".